# Additive Manufactured Polymer-Bonded Isotropic NdFeB Magnets by Stereolithography and Their Comparison to Fused Filament Fabricated and Selective Laser Sintered Magnets

**DOI:** 10.3390/ma13081916

**Published:** 2020-04-19

**Authors:** Christian Huber, Gerald Mitteramskogler, Michael Goertler, Iulian Teliban, Martin Groenefeld, Dieter Suess

**Affiliations:** 1Physics of Functional Materials, University of Vienna, 1090 Vienna, Austria; dieter.suess@univie.ac.at; 2Christian Doppler Laboratory for Advanced Magnetic Sensing and Materials, 1090 Vienna, Austria; 3Incus GmbH, 1220 Vienna, Austria; gmitteramskogler@lithoz.com; 4Institute for Surface Technologies and Photonics, Joanneum Research Forschungsgesellschaft GmbH, 8712 Niklasdorf, Austria; michael.goertler@joanneum.at; 5Magnetfabrik Bonn GmbH, 53119 Bonn, Germany; Iulian.Teliban@Magnetfabrik.de (I.T.); Martin.Groenefeld@magnetfabrik.de (M.G.)

**Keywords:** magnets, photopolymerization, powder bed fusion, material extrusion, NdFeB

## Abstract

Magnetic isotropic NdFeB powder with a spherical morphology is used to 3D print magnets by stereolithography (SLA). Complex magnets with small feature sizes in a superior surface quality can be printed with SLA. The magnetic properties of the 3D printed bonded magnets are investigated and compared with magnets manufactured by fused filament fabrication (FFF), and selective laser sintering (SLS). All methods use the same hard magnetic isotropic NdFeB powder material. FFF and SLA use a polymer matrix material as binder, SLS sinters the powder directly. SLA can print magnets with a remanence of 388 mT and a coercivity of 0.923 T. A complex magnetic design for speed wheel sensing applications is presented and printed with all methods.

## 1. Introduction

Additive manufacturing (AM) offers a new era of possibilities for magnetic materials and advanced magnetic sensing applications. AM methods creating solid structures layer-by-layer from a formless or form-neutral feedstock by means of chemical or thermal processes. This leads to several advantages such as design freedom, net shape capabilities, waste reduction and minimum lead times for prototyping, compared to traditional manufacturing methods like sintering of full-dense magnets or injection-molding of polymer-bonded magnets.

Fused filament fabrication (FFF) or fused deposition modeling (FDM) is a well-known and widely used AM method to print thermoplastic materials. It uses a wire-shaped thermoplastic filament as building material. The filament is feed to a movable extruder where it heated up above its softening point. The molten material is pressed out of the extruder nozzle and builds the structure layer-by-layer on the already printed and solidified layer [1]. A sketch of the FFF method is shown in Figure 1a. By mixing magnetic soft- or hard magnetic materials into the thermoplastic binder, FFF can be also used to 3D print polymer-bonded magnets with filling ratios up to 90 wt.%. [2,3,4,5,6,7,8,9]. A big disadvantage of polymer-bonded permanent magnets is their lowered maximum energy product ${\left(BH\right)}_{\max}$ compared to sintered magnets due to their plastic matrix material.

To maximize the performance of permanent magnets, the ${\left(BH\right)}_{\max}$ must be increased. Powder bed fusion (PBF) processes does not need a polmyer matrix material. PBF uses a high-power laser or electron beam source to sinter or completely melt the metallic powder [10]. To optimize the printing process and the quality of the prints, powders with a spherical morphology are preferred [11]. Figure 1b shows a sketch of the printing process. This means theoretically, that fully dense magnets can be printed. However, the rapid liquefaction and cooling rates of the material at the localized heat source, influences the microstructure (size of the grains and the composition of the grain boundaries), and therefore the magnetic properties of the printed structures [12,13,14]. The optimization of the magnetic properties of PBF processed magnets is an active research field. The PBF process can be divided into the heating source. Following commonly used PBF printing techniques are used to investigate the capability to print hard or soft magnets: electron beam melting (EBM) [15], selective laser melting (SLM) [16,17,18,19,20,21], and selective laser sintering (SLS). SLS does not completely melt each powder layer but sinter the particles to retain their original microstructure. As a second step, the coercivity of the SLS printed samples can be substantially increased by a grain boundary infiltration method [22].

Stereolithography (SLA) was the first commercially available AM technology. The layers of the sliced computer model is scanned by a visible or ultraviolet (UV) light to cure the photosensitive resin selectively for each cross-section, or a digital light processing (DLP) engine projects and cure the whole image of every layer. After each finished layer, the workpiece is lowered by one-layer thickness. Then, the resin sweeps across the cross-section of the partly finished object, and coating it with a new layer of fresh resin. This layer is scanned and cured-on the previous hardened layer. Figure 1c shows the principle of SLA. SLA of soft magnetic materials with a very low filler content of only 30 wt.% is described in Reference [23]. Up to now, no publication about SLA of hard magnetic materials exists.

This publication deals with SLA of magnetic isotropic powder in a photo reactive resin. The resolution and quality of the 3D printed permanent magnetic samples are superior. Figure 1d shows a 3D printed magnetic St. Stephen’s Cathedral, Vienna with a minimum feature size of the model of 0.1 mm and a layer height of 60 µm. Furthermore, the same magnetic isotropic powder is used to print polymer-bonded magnets with FFF and sintered magnets with SLS as described in our prior works [2,16]. All advantages and disadvantages of each method are discussed in detail. Complex magnets are printed and their magnetic properties are investigated and compared.

## 2. Materials & Methods

We used a commercial isotropic NdFeB powder (MQP-S-11-9 from Magnequench Corporation) for all three presented AM methods. This powder has a spherical morphology with a powder size distribution of $d_{50}$ of 38 µm and the tap density exhibits 61% of the materials full density. A scanning electron microscopy (SEM) image of the powder can be seen in Figure 2a. Its main field of application is the manufacturing of bonded magnets, particularly by injection molding or extrusion. The powder particles have nano-sized NdFeB grains, it have a uniaxial magnetocrystalline anisotropy which are random orientated. This leads to magnetic isotropic behavior of the bulk magnet. This powder is produced by a gas atomization process and a followed heat treatment. The chemical composition states ${\mathrm{Nd}}_{7.5}$${\mathrm{Pr}}_{0.7}$${\mathrm{Fe}}_{75.4}$${\mathrm{Co}}_{2.5}$$B_{8.8}$${\mathrm{Zr}}_{2.6}$${\mathrm{Ti}}_{2.5}$ (at.%) [20].

Incus GmbH developed an industrial vat photopolymerisation process called Lithography-based Metal Manufacturing (LMM). The LMM machine is based on a top-down SLA principle. The liquid photo-reactive feedstock is polymerized from above by a high-performance projection unit (Figure 1c). The building platform with the submerged parts is lowered, layer-by-layer, according to the chosen layer thickness. For this study, a layer thickness of 60 µm is used. After the curing of a layer, the wiper blade applies a fresh film of feedstock. The size of the building platform is 75×43
${\mathrm{mm}}^{2}$ and the resolutions in the *x* and *y* directions are 40 µm each. The printing time of a single layer is 35 s, which results in a build speed of 6 mm/h in *z*-direction (about 20 ${\mathrm{cm}}^{3}$/h in volume). A photo-reactive feedstock is prepared, based on commercially available di- and polyfunctional methacrylates (60 wt.%). The reactive components included an initiation system and a proprietary photoinitiator, which absorbs light in the wavelengths emitted by the projector. A solid loading of MQP-S powder up to 92 wt.% is achieved. The binder components and the magnetic powder were added in a mixing cup and homogeneously dispersed via centrifugal mixing. The self-supporting function of the material facilitates the volume-optimized placement of different parts on a single building platform without the need for additional support structures. Typically, the LMM printed component is sintered after the printing process to create a full dense structure. In our case, the LMM process is used without sintering, therefore polymer-bonded magnets are created.

For the FFF of bonded magnets, a conventional end-user 3D printer Builder from Code P is used as described in Reference [2]. We are using a prefabricated compound (Neofer ® 25/60p) from Magnetfabrik Bonn GmbH. It consists of 89 wt.% (52 vol.%) MQP-S powder inside a PA11 matrix. The magnetic compound has a remanence of $B_{r}=387$ mT and a coercivity of $H_{cj}=771$ kA/m. To get the wire-shaped filaments for the 3D printer extruder, the Neofer ® 25/60p compound granules are extruded at the University of Leoben with a Leistritz ZSE 18 HPe-48D twin-screw extruder. The extrusion temperature is 260 °C, and the hot filament is hauled off and cooled by a cooled conveyor belt. The diameter of 1.75 mm and tolerances of the filament are controlled by a Sikora Laser Series 2000 diameter-measuring system. The Builder 3D printer can build structures with a maximum size of 220×210×165
${\mathrm{mm}}^{3}$ (L × W × H). The layer height resolution can be varied between 0.05 and 0.5 mm. Printing speed ranges from 10 to 80 mm/s, traveling speed ranges from 10 to 200 mm/s. To avoid clogging of the nozzle due to the height filler content, the minimum nozzle size diameter is 0.4 mm. This large nozzle diameter defines the minimum feature size of the prints. The printing temperature for the PA11 compound is 260 °C. For a better adhesion of the first layer, the printing bed is heated up to 80 °C.

For sample fabrication with the SLS system, a commercial Farsoon FS121M LPBF-machine is used [16]. It is equipped with a continuous wave 200 W Yb-fibre laser with a wavelength of 1.07 µm and a spot size of 0.1 mm. It has a build space of 120×120×100
${\mathrm{mm}}^{3}$ (L × W × H). The printing of the MQP-S powder is performed under Ar atmosphere with oxygen content below 0.1%. A layer thickness of 100 µm, and the powder recoating was done with a carbon fiber brush. All specimens were printed without support structures directly onto a steel substrate plate to ensure proper heat dissipation. The laser power *P* is varied between 20 W and 100 W, and the scan speed *v* is varied between 50 mm/s and 2000 mm/s to find the optimal printing parameters. The line energy $E_{\mathrm{line}}=P/v$ is a convenient printing parameter. For sintering of the MQP-S powder, line energies between 0.03 J/mm and 0.07 J/mm at 40 W, and a hatch spacing *h* of 0.14 mm is practicable.

## 3. Results & Discussion

The focus in this paper is the discussion of the magnetic properties of the different used AM methods. To test the magnetic properties, cubes with a dimensions of 5×5×5
${\mathrm{mm}}^{3}$ are printed with the above described printing parameters and techniques. No post-processing of the printed samples is performed. SEM images of the surface and the layer structure are presented in Figure 2b–d). The sample printed by FFF shows the rawest surface (Figure 2b). Figure 2b indicating a partly densified SLS sample while several cracks can still be seen in the microstructure. The surface of the SLA sample shows the best quality of all three methods (Figure 2d).

Volumetric mass density ϱ is measured with with a hydrostatic balance (Mettler Toledo, AG204DR) based on the Archimedes principle. The filling fraction of the MQP-S powder inside the polymer matrix for the FFF and the SLA printed samples is measured with by loss on ignition (LOI), whereby the sample is heated up to 1100 °C [24] (Table 1). All measurements are performed with three different samples to minimize the statistical error. The density for the FFF sample is around 20% lower compared to the theoretical value (ϱ=4.35
${\mathrm{g/cm}}^{3}$). SLS shows a density that is in the same range as the tap density of the powder (ϱ=4.3
${\mathrm{g/cm}}^{3}$). This shows that the MPQ-S powder is sintered without complete melting of the material. SLA has the highest volumetric mass density of the investigated printing techniques. For the measurement of the magnetic hysteresis curve and the magnetic properties of the samples, a permagraph (magnetic closed loop measurement) from Magnet-Physik Dr. Steingroever GmbH with a JH 15-1 pick-up coil is used. The magnetic hysteresis curved are shown in Figure 3, and the magnetic properties are summarized in Table 1.

The coercivity of the isotropic MQP-S NdFeB powder depends only on the microstructure of the material and it is unchanged for the FFF and SLA process. For SLS parts, the coercivity is around 25% lower compared to the data sheet value of the powder. This is a result of the inhomogeneous microstructure, in particular the grain size distribution [22].

The capabilities of the different presented AM methods are discussed on a magnetic speed wheel sensing system. Such high precision sensor systems are embedded in many applications, especially in automotive application, for example, in anti-blocking system (ABS) or engine management systems [25]. A possible design of such speed sensors consist of a magnetic field sensor, for example, Hall effect or giant magnetoresistance (GMR) sensor, a permanent magnet which provide a bias field and a soft magnetic wheel. Normally, the magnet is underneath the sensor (back-bias magnet) and the rotating soft magnetic wheel modulates the magnetic field of the back-bias magnet. The rotational velocity of the wheel is direct proportional to the modulation of the field. Figure 4a shows a sketch of a possible wheel speed sensing arrangement.

If a GMR sensor is designated to detect the field modulation, some special magnetic design criteria must be considered. GMR sensors are in-plane sensitive and the linear range is very small [26]. This means that the back-bias magnet must have very low magnetic in-plane field components. This can be achieved by a specific geometric design of the magnet. Finite element simulation methods (FEM) can optimize the topology of a magnet for tailored field distributions [4,27]. For magnetic isotropic materials, the geometric design is the only way to define the magnetic flux density distribution outside the magnet. The remanence $B_{r}$ of the material is proportional to the filling fraction of bonded magnets, and it determines the magnitude of the flux density. Figure 4b shows the cross-section of a well-known geometry that minimizes the components of the magnetic stray field B in *x* and *y* direction in a wide range along the *x*-axis $r_{x}$. In this case, an accurate field distribution is more important than a maximum field. Prototyping of such complex magnetic designs is one of the biggest advantage of AM methods. For the mass production of such bonded magnets, injection molding is the preferred manufactured method.

Starting from the design as described above, a back-bias magnet for speed wheel sensing is 3D printed with: (i) FFF, (ii) SLS, and (iii) SLA. The overall size of the magnet is 7×5×5.5
${\mathrm{mm}}^{3}$ (L × W × H). After the printing process, the magnet is magnetized in an electromagnet with a maximum magnetic flux density of 1.9 T in permanent operation mode. Figure 5 shows a line scan of the magnetic flux density B, 2.5 mm above the pyramid tip (T) for all three AM methods. The magnetic flux density is measured with a Hall probe and the FFF 3D printer as described in Reference [2]. The magnet printed by FFF has the weakest flux density $B_{z}$ because of the smaller remanence $B_{r}$ compared to the other AM methods. The remanence is directly proportional to the volumetric mass density ϱ and filling fraction $w_{f}$, respectively [3]. Magnets produced by SLA has the highest density and therefore the maximum magnitude of the magnetic flux density. Higher filling loads of the filaments for FFF would increase the remanence and therefore the magnitude of the flux density, but this would lead to very brittle filaments which are not processable with commercial available FFF extruders. However, all three methods show a minimum stray field $B_{x}$ and $B_{y}$ along the *x*-axis. A picture of the printed magnets is illustrated in Figure 6. It is clearly visible that SLA produces the geometrical most accurate prints.

## 4. Conclusions

AM of magnetic materials with different methods and materials is an active research field. Many groups use the hard magnetic isotropic NdFeB MQP-S powder due to the spherical morphology and robustness against corrosion.

Nevertheless, this publication describes SLA of hard magnetic materials for the first time. Even more, the SLA printing method of bonded NdFeB magnets is compared to previous published results of FFF and sintered magnets printed with SLS. Magnets printed with SLA show the best magnetic performance and a very high surface quality compared to samples printed with FFF or SLS. The modification of the microstructure of the powder during the SLS process is the reason for its lower magnetic performance compared to the other methods. FFF is the most affordable and simplest way to print magnets, but due to the large nozzle diameter, the accuracy of the physical dimensions is limited. Additionally, the lower volumetric mass density compared to the theoretical value is a reason for the lower remanence of the printed magnets.

In summary, it can be said that the MQP-S powder perfectly meets the requirements of the SLA printing process. We can see a huge potential for the manufacturing of complex magnetic designs in a superior quality.

## Figures and Tables

**Figure 1 materials-13-01916-f001:**
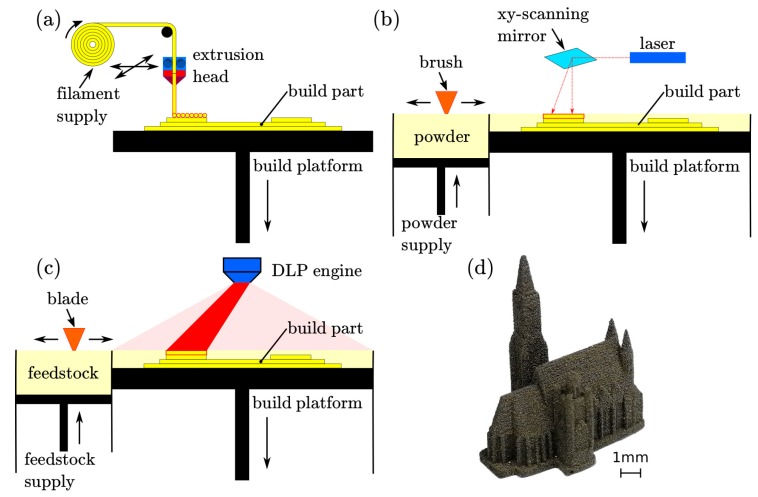
Different used additive manufacturing (AM) methods. (**a**) fused filament fabrication (FFF). (**b**) selective laser sintering (SLS). (**c**) stereolithography (SLA). (**d**) 3D printed magnetic St. Stephen’s Cathedral, Vienna by SLA.

**Figure 2 materials-13-01916-f002:**
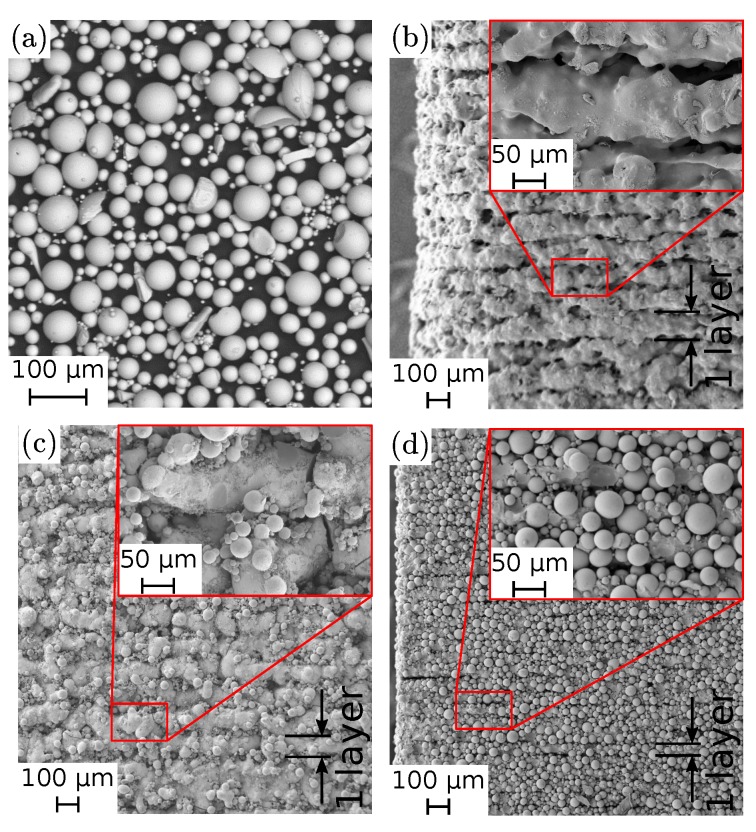
All presented AM methods use the same isotropic NdFeB powder (MQP-S-11-9, Magnequench). (**a**) scanning electron microscope (SEM) image of the initial MQP-S powder. SEM images of the surfaces of magnetic samples, printed with: (**b**) FFF, (**c**) SLS, (**d**) SLA.

**Figure 3 materials-13-01916-f003:**
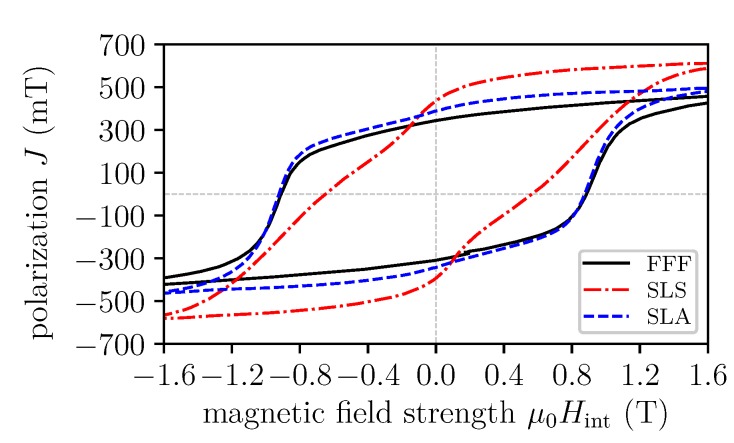
Hysteresis loops of the different printing methods.

**Figure 4 materials-13-01916-f004:**
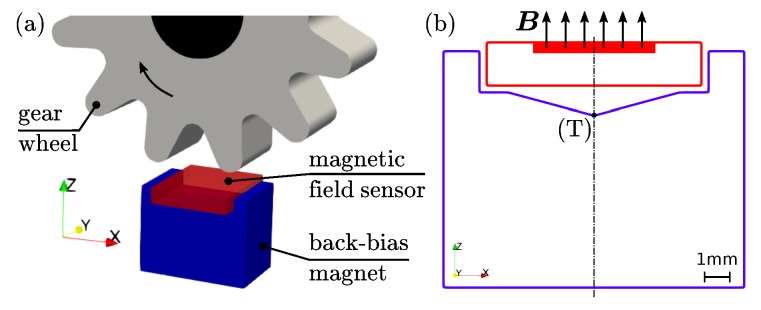
Magnetic wheel speed sensing. (**a**) Principle of the magnetic speed sensing. A permanent magnet is underneath the magnetic field sensor (back-bias magnet). A soft magnetic gear periodically modulates the bias field of the magnet. (**b**) Special back-bias magnet design for giant magnetoresistance (GMR) sensors.

**Figure 5 materials-13-01916-f005:**
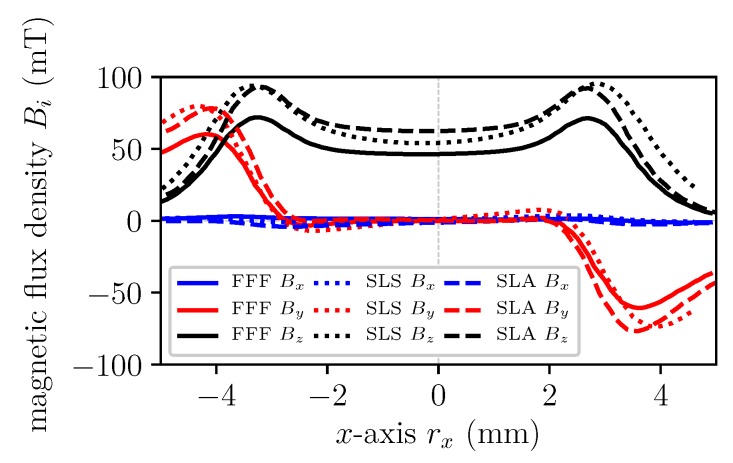
Line scan of the magnetic flux density B, 2.5 mm above the pyramid tip (T).

**Figure 6 materials-13-01916-f006:**
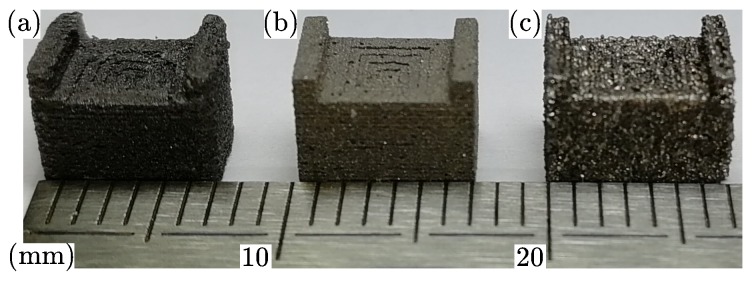
Picture of the back-bias magnets printed by: (**a**) FFF, (**b**) SLA, (**c**) SLS.

**Table 1 materials-13-01916-t001:** Properties of the isotropic NdFeB powder (MQP-S-11-9 from Magnequench Corporation.) and the samples printed with the different AM methods. $w_{f}\ldots$ filling mass fraction, ϱ… volumetric mass density, $B_{r}\ldots$ residual Induction, and $\mu_{0}H_{\mathrm{cj}}\ldots$ intrinsic coercivity.

Sample	$w_{f}$ (wt.%)	ϱ ${\mathrm{(g/cm}}^{3}$)	$B_{r}$ (mT)	$\mu_{0}H_{\mathrm{cj}}$ (T)
powder	–	7.43	746	0.880
FFF	89	3.57	344	0.918
SLS	100	4.47	436	0.653
SLA	92	4.83	388	0.923

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
