# Peer review of "Additive Manufactured Polymer-Bonded Isotropic NdFeB Magnets by Stereolithography and Their Comparison to Fused Filament Fabricated and Selective Laser Sintered Magnets"

_materials, 2020, doi:10.3390/ma13081916_

Round 1
Reviewer 1 Report
This is an interesting work. Authors compared the microstructures and magnetic properties of 3D print magnets in detail which were printed with three different additive manufacturing methods. The results of SEM and magnetic measurements demonstrate advantages and disadvantages of each method, and highlight the high quality of magnet printed via SLA method, as shown with the best magnetic performance and a very high surface quality. In addition, the design of magnetic speed wheel sensing system is interesting and it reveals the maximum magnitude of the magnetic flux density present in magnet printed via SLA method in high precision. Therefore, this work provides a promising route for optimizing the manufacturing of complex magnetic designs via SLA and I believe that it will attract the interest of many researchers working in the field. I am thus pleased to recommend the acceptance of this work in Materials.
Author Response
This is an interesting work. Authors compared the microstructures and magnetic properties of 3D print magnets in detail which were printed with three different additive manufacturing methods. The results of SEM and magnetic measurements demonstrate advantages and disadvantages of each method, and highlight the high quality of magnet printed via SLA method, as shown with the best magnetic performance and a very high surface quality. In addition, the design of magnetic speed wheel sensing system is interesting and it reveals the maximum magnitude of the magnetic flux density present in magnet printed via SLA method in high precision. Therefore, this work provides a promising route for optimizing the manufacturing of complex magnetic designs via SLA and I believe that it will attract the interest of many researchers working in the field. I am thus pleased to recommend the acceptance of this work in Materials.
We thank the reviewer for the acceptance of the paper.
Reviewer 2 Report
In the presented manuscript: “Additive manufactured polymer-bonded isotropic NdFeB magnets by stereolithography and their comparison to fused filament fabricated and powder bed fused magnets” by Christian Huber et al., the authors describe the NdFeB powder used to 3D print magnets in stereolithography (SLA). The subject of submitted paper is in general interesting but some more proper experiments are needed.
- Why only one set of experiments is presented.
- There is any technical dependence on print quality?
- How morphology of print is related to its final magnetic properties.
Author Response
In the presented manuscript: “Additive manufactured polymer-bonded isotropic NdFeB magnets by stereolithography and their comparison to fused filament fabricated and powder bed fused magnets” by Christian Huber et al., the authors describe the NdFeB powder used to 3D print magnets in stereolithography (SLA). The subject of submitted paper is in general interesting but some more proper experiments are needed.
1. Why only one set of experiments is presented.
The main focus of this publication is the comparison of the different printing techniques to 3D print magnetic structures. We have shown that SLA provides complex geometries with a superior surface quality (Fig. 1 (d)). The wheel speed sensing example should demonstrate the effectiveness of all three methods (Fig. 5), but SLA provides the best quality of the surface and the geometric specifications.
2. There is any technical dependence on print quality?
As discussed in section 2 and Fig. 2, SLA provides the best printing results of all used methods.
3. How morphology of print is related to its final magnetic properties.
The magnetic properties of the final magnet printed by SLA and FFF depend only on the filling ratio of the magnetic particles. For SLS, the micro-structure and the grain-size of the printed structure influences the magnetic properties.
Reviewer 3 Report
The paper deals with the additive Layer Manufacturing of NdFeB magnets with different techniques. The topic is really interesting and novel.
The methodology section is too brief and lacks detail. Also, some methods are not described but are used later on. Some methodology is found in the results section, some results in the methods section. The results on the whole do not contain assessment of errors.
For instance SEM images should be in the results, but they are reported in methods section. Methods section doesn't include description of SEM analysis and instruments have been used. No details of TGA analysis are reported and no figures. How residual mass has been calculated? Printing process parameters are given with lack of details (range of values).
Author Response
The paper deals with the additive Layer Manufacturing of NdFeB magnets with different techniques. The topic is really interesting and novel.
The methodology section is too brief and lacks detail. Also, some methods are not described but are used later on. Some methodology is found in the results section, some results in the methods section. The results on the whole do not contain assessment of errors.
For instance SEM images should be in the results, but they are reported in methods section. Methods section doesn't include description of SEM analysis and instruments have been used. No details of TGA analysis are reported and no figures. How residual mass has been calculated? Printing process parameters are given with lack of details (range of values).
We thank the referee for very important suggestions and pointing out inconsistencies in the manuscript. We rearranged several parts of the manuscript.
There was error of the filling fraction measurements, in this publication we used a loss on ignition (LOI) measurement for the determination of the filling fraction and the residual mass. We reformulated the sentence.
We discussed the used printing parameter of all methods. It was not the scope of this publication to perform detailed parameter studies to find the best and optimum printing parameters.
Reviewer 4 Report
The authors designed and manufactured 3D print magnets with small feature sizes and discussed their magnetic properties. This paper is an important contribution but there are some unclear points as follows:
1) There are many abbreviated names in this manuscript and they sometimes confuse readers. Is there any difference between FFF and FDM ? In abstract, the authors investigate the magnetic properties of 3D print magnets by SLA and that compared with those by FFF and PBF. However, the authors describe the hysteresis loop and magnetic flux destiny of the magnets made by FDM, SLS, and SLA in figures 3 and 5 although FDM is not described in the text in section 2 and 3 at all.
2) How about PBF? The reviewer understood that PBF printing technique includes EBM, SLM, and SLS. However, the authors discuss only about SLS in all figures and the text. Indeed, in the section of conclusion, the authors describe the superiority of SLA compared with FFF and SLS. Why the author discuss about PBF instead of SLS only in abstract?
Author Response
The authors designed and manufactured 3D print magnets with small feature sizes and discussed their magnetic properties. This paper is an important contribution but there are some unclear points as follows:
1) There are many abbreviated names in this manuscript and they sometimes confuse readers. Is there any difference between FFF and FDM ? In abstract, the authors investigate the magnetic properties of 3D print magnets by SLA and that compared with those by FFF and PBF. However, the authors describe the hysteresis loop and magnetic flux destiny of the magnets made by FDM, SLS, and SLA in figures 3 and 5 although FDM is not described in the text in section 2 and 3 at all.
We thank the referee for this hint. We changed the labels of the figures.
2) How about PBF? The reviewer understood that PBF printing technique includes EBM, SLM, and SLS. However, the authors discuss only about SLS in all figures and the text. Indeed, in the section of conclusion, the authors describe the superiority of SLA compared with FFF and SLS. Why the author discuss about PBF instead of SLS only in abstract?
We changed the title of the publication from “Additive manufactured polymer-bonded isotropic NdFeB magnets by stereolithography and their comparison to fused filament fabricated and powder bed fused magnets” to “Additive manufactured polymer-bonded isotropic NdFeB magnets by stereolithography and their comparison to fused filament fabricated and selective laser sintered magnets” and we changed PBF to SLS in the abstract.
Round 2
Reviewer 3 Report
The article has been properly revised, the content is novel and of high scientific interest. The authors clarified the limits and potentiality of the diffentent AM of hard magnets in terms of magnetic performance and surface finishing.